# Mitochondrial DNA Changes in Respiratory Complex I Genes in Brain Gliomas

**DOI:** 10.3390/biomedicines11041183

**Published:** 2023-04-15

**Authors:** Paulina Kozakiewicz, Ludmiła Grzybowska-Szatkowska, Marzanna Ciesielka, Paulina Całka, Jacek Osuchowski, Paweł Szmygin, Bożena Jarosz, Marta Ostrowska-Leśko, Jarosław Dudka, Angelika Tkaczyk-Wlizło, Brygida Ślaska

**Affiliations:** 1Department of Radiotherapy, Oncology Centre of Lublin St. Jana z Dukli Jaczewskiego 7, 20-090 Lublin, Poland; 2Department of Radiotherapy, Medical University in Lublin, Chodźki 7, 20-093 Lublin, Poland; 3Chair and Department of Forensic Medicine, Medical University in Lublin, Jaczewskiego 8b, 20-090 Lublin, Poland; 4Chair and Department of Neurosurgery and Pediatric Neurosurgery, Medical University in Lublin, Jaczewskiego 8, 20-090 Lublin, Poland; 5Chair and Department of Toxicology, Medical University in Lublin, Jaczewskiego 8b, 20-090 Lublin, Poland; 6Institute of Biological Bases of Animal Production, University of Life Sciences in Lublin, Akademicka 13, 20-950 Lublin, Poland

**Keywords:** mitochondria, brain tumor, mtDNA polymorphisms, ND1

## Abstract

Mitochondria are organelles necessary for oxidative phosphorylation. The interest in the role of mitochondria in the process of carcinogenesis results from the fact that a respiratory deficit is found in dividing cells, especially in cells with accelerated proliferation. The study included tumor and blood material from 30 patients diagnosed with glioma grade II, III and IV according to WHO (World Health Organization). DNA was isolated from the collected material and next-generation sequencing was performed on the MiSeqFGx apparatus (Illumina). The study searched for a possible relationship between the occurrence of specific mitochondrial DNA polymorphisms in the respiratory complex I genes and brain gliomas of grade II, III and IV. The impact of missense changes on the biochemical properties, structure and functioning of the encoded protein, as well as their potential harmfulness, were assessed in silico along with their belonging to a given mitochondrial subgroup. The A3505G, C3992T, A4024G, T4216C, G5046A, G7444A, T11253C, G12406A and G13604C polymorphisms were assessed as deleterious changes in silico, indicating their association with carcinogenesis.

## 1. Introduction

The mitochondrion is an organelle that is an important energy transformer in eukaryotic cells, taking part in the process of cellular oxidation. As a result of electron transport through the respiratory chain, oxidative phosphorylation and ATP synthesis occurs in the mitochondria [1]. The four protein complexes located in the inner mitochondrial membrane and ATP synthase are called the fifth complex; they create the respiratory chain. The transporters are arranged according to the increasing oxyreduction potential, which ensures high efficiency of energy generation. The first four complexes transfer electrons to the oxygen molecule. Complex I is named reduced nicotinamide adenine dinucleotide (NADH)–ubiquinone; complex II is succinate–ubiquinone oxidoreductase, otherwise described as succinate dehydrogenase (SDH); complex III is the cytochrome *bc*_1_ complex; complex IV is described as cytochrome c oxidase (COX) [1,2]. The first complex consists of as many as 45 subunits and oxidizes NADH by transferring two electrons from NADH to ubiquinone (Q), giving ubiquinol (OH_2_) (NADH + Q + 5H^+^_matrix_ → NAD^+^ + QH_2_ + 4H^+^_cytosol_). The transport of electrons to ubiquinone in complex I includes additional transporters, such as flavin mononucleotide (FMN) and iron–sulfur (Fe-S) centers. The transfer of two electrons from NADH to ubiquinone is accompanied by the transfer of four protons from the mitochondrial matrix through the inner mitochondrial membrane into the intermembrane space. Seven out of forty-five subunits of complex I are encoded by mitochondrial DNA (*ND1*, *ND2*, *ND3*, *ND4*, *ND4L*, *ND5*, *ND6*), while the left-over 38 are encoded by nuclear DNA [2]. The largest component of the mitochondrial respiratory chain is complex I. Its malfunction has the greatest impact on the initiation of disorders related to the disruption of oxidative phosphorylation [3]. It is the first link in the mitochondrial respiratory chain transferring electrons from NADH to ubiquinone and pumping protons outside the inner mitochondrial membrane [1]. There is a relationship between the event of changes in mitochondrial DNA and a higher risk of cancer (breast cancer, prostate cancer, melanoma, rectal cancer, stomach cancer, endometrial cancer, bladder cancer, ovarian cancer) and chronic diseases (Leber’s hereditary optic neuropathy (LHON), hypertension, metabolic syndrome, bipolar disorder, coronary artery disease, diabetes type 2) [4]. In patients with LHON, the most frequently described somatic mutations in the MITOMAP database involve the *ND1* and *ND6* regions, and these are G3460A (A52T) and T14484C (M64V), respectively [5]. It has been shown that in patients (62), following radical nephrectomy due to renal cancer (mostly clear cell cancer) with *ND1* mutations present in the tumor, the 5-year relapse free time (RFS) (*p* = 0.0006) was shorter. The mutations were mostly heteroplasmic (84.2%). There are reports that the mutation 3572.1C, affecting the *ND1* region described in the cited work, is related to oncocytic tumors with eosinophilic cytoplasm (oncocytic pituitary adenoma, oncocytic thyroid carcinoma or benign renal oncocytoma) [6]. The granular eosinophilic cytoplasm of these cells results from the accumulation of a large number of mitochondria [7]. In haplogroup J individuals of the European population, an increased risk of LHON syndrome has also been reported, which is related to the 14484T>C polymorphism in ND6 [8]. In the literature, the missense change T14634C (M14V) in *ND6* has been reported in hypoxia-resistant glioblastoma cell lines and as a variant altering the spatial structure of the protein [9].

In the material studied, the changes in the genes of complex I were assessed in terms of the polymorphisms present and their belonging to a given haplogroup, and in silico, the possible impact of missense changes on the functioning of proteins were assessed.

The Bioethics Committee at the Medical University of Lublin approved the research included in this study (number KE-0254/171/2017).

## 2. Methodology

### 2.1. Material

The material used for the research consisted of blood and fragments of tumor tissue collected during brain tumor removal surgeries at the Department of Neurosurgery and Pediatric Neurosurgery at the Independent Public Clinical Hospital No. 4 in Lublin in 2016–2019. The group size was 30 patients. The study included material from patients diagnosed with grade II, III and IV glioma. Due to the new WHO (World Health Organization) classification from 2016, which takes into account the tumor genotype, the mutation status of *IDH* (isocitrate dehydrogenase) genes in the collected tumor tissue fragments was additionally determined. Changes affecting codon 132 of the *IDH1* gene and codon 172 of the *IDH2* gene were examined using Sanger sequencing. The patients with brain tumors received no previous oncological treatment. They ranged in age from 22 to 76 years, with a mean age of 50.2 years. Patient characteristics are presented in Table 1.

### 2.2. Methods

DNA was isolated from sections of brain tumors and peripheral blood using a DNA isolation kit (DNeasy Blood and Tissue Kit, Qiagen, Hilden, Germany), in accordance with the manufacturer’s instructions. The concentration and the degree of purity of the obtained DNA were evaluated by spectrophotometry using a NanoDrop 1000 spectrophotometer (Thermo Scientific). The sequencing procedure was carried out using the Nextera XT DNA Library Preparation Kit and the Nextera XT Index Kit (Illumina), and it included the following steps: amplification of the entire mtDNA genome, purification of the obtained products and assessment of their concentration, preparation of libraries and high-throughput sequencing carried out on the MiSeq device (Illumina) with a chemistry-based kit—MiSeq Reagent Kit v. 3 (600 cycles). The integrated sequencer software—MiSeq Control Software, Real-Time Analysis Software, MiSeq Reporter—made it possible to run and control the process and then generate files in FASTQ format.

The sequenced genomes were compared with the reference sequence of the human mtDNA (GenBank NC_012920, AC_000021) using the mtDNA Variant Processor and mtDNA Variant Analyzer software available on the Base Space Illumina platform. The changes obtained from the neoplastic tissue were compared with the variants from the blood of the respective patient. A change occurring in mtDNA in both the blood and the tumor of the patient was identified as a polymorphism (germ line change). An mtDNA change occurring only in the cells of the tumor tissue was identified as a mutation (somatic change). The affiliation of mtDNA polymorphisms to a given mitochondrial subgroup was established using the PhyloTree.org–mtDNA tree Build 17 function available at https://www.mitomap.org/MITOMAP (accessed on 5 August 2022).

Missense changes were analyzed in silico by using the following bioinformatics programs:ExPASy Proteomics tools, ProtParamProgram—assessment of the biochemical properties of the protein such as theoretical isoelectric point (pI), protein stability index, aliphatic index and grand average of hydropathicity (https://www.expasy.org/resources/protparam) [10];Pfam version 33.1—assessment of the secondary protein structure, including the number of helics (http://pfam.xfam.org/search#tabview=tab1) [11];AGADIR—theoretical prediction of the percentage of the protein helix (http://agadir.crg.es/protected/academic/calculation4.jsp) [12];TMHMM 2.0 program—prediction of transmembrane helices in a protein (http://www.cbs.dtu.dk/services/TMHMM/) [13];PSSM Viewer—position specifics score matrices (http://www.ncbi.nlm.nih.gov/Class/Structure/pssm/pssm_viewer) [14];SIFT Sequence—predicting the effect of amino acid substitution on protein function Score: tolerated change > 0.05, change affecting protein function ≤ 0.05 (https://sift.bii.a-star.edu.sg/www/SIFT_seq_submit2.html) [15];MitImpact 3D-predictor APOGGE, mitochondrial database—assessment of the potential pathogenicity of amino acid substitutions. Score: neutral change ≤ 0.5, pathogenic change > 0.5 (https://mitimpact.css-mendel.it/) [16];ConSurf-Database—analysis of evolutionary conservation profiles for proteins. Score on a 1–9 scale (1–3 variable region, 4–6 moderately conserved region, 7–9 highly conserved region) and normalized points (variable region < 0, moderately conserved region 0–0.5, highly conserved region > 0.5) (https://consurfdb.tau.ac.il/index.php) [17,18].

## 3. Results

Polymorphisms results for *ND1*, *ND2*, *ND3*, *ND4*, *ND4L*, *ND5* and *ND6* are presented in Table 2. The assessment of the impact of polymorphisms causing amino acid changes on biochemical properties and the probability of affecting protein functioning are presented in Appendix A.

In the mitochondrial complex I in the examined material extracted from 30 patients, 21 missense and 46 sense various polymorphisms were found (67 total). The most frequent sense-change polymorphism was T4216C, which affected 13.3(3)% of patients (4/30). The second most frequent changes were G13145A and A10398G. The transitions of A10398G and G13145A occurred in a total of 10% of patients, with grade III (2/30) and II (1/30) gliomas, respectively. T4216C polymorphism occurred in two patients with grade IV glioma, one with grade II and one with grade III. The most common polymorphisms were found in *ND2* (45 times), while the most types were found in *ND5* (18). Two changes found (G13604C, A13866G) have not been reported in the literature yet. 

### 3.1. Polymorphisms in the ND1 Gene

In total, nine types of polymorphism were found in *ND1* (Table 1), including as many as six missenses (T4216C, A3505G, C3992T, A4024G, A4093G, A3796G). Among the polymorphisms causing amino acid change, the T4216C (Y304H) polymorphism had the greatest impact on the biochemical values of the protein and, as mentioned above, it occurred in 13.3(3)% of patients (4/30). The frequency of this change, according to the mtDB-Human Mitochondrial Genome Database, turned out to be quite high, which translated into the obtained results (Table 2). The second most frequent missense polymorphism in the ND1 subunit gene was A3505G (T67A), which occurred in 6.6(6)% of patients (2/30) (Table 2). Another of the missense polymorphisms that occurred in patient with grade IV tumor was C3992T (T229M), (Table 2). The missense polymorphism A4024G (T240A), detected in one patient with glioblastoma multiforme diagnosis, did not significantly affect the protein properties (Appendix A). The missense polymorphism A4093G (T263A), similar to the previously described variant, was detected in one patient with IV tumor diagnosis (Table 2). The frequency of this polymorphism, according to the mtDB-Human Mitochondrial Genome Database, turned out to be very rare (2) (Appendix A). Each of the three detected synonymous polymorphisms (A3480G, G3915A, T3826C) occurred in individual patients. It is worth noting that the analysis of the A3505G (T67A) polymorphism in the TMHMM program showed a shift of 5fiveamino acids (positions 62–67) from the mitochondrial matrix to the transmembrane segment, which, in turn, resulted in an increase in the length of the extra-mitochondrial segment (Appendix A Appendix A).

### 3.2. Polymorphisms in the ND2 Gene

In total, twelve types of polymorphisms were found in *ND2*, including two sense changes—G5460A and G5046A. The A4769G synonymous change occurred in all patients (30/30) (Table 2). The missense polymorphisms of the *ND2* gene sequences resulted in slight deviations in the protein biochemical values (Appendix A). The G5460A (A331T) polymorphism occurred at a frequency of 13.3(3)% (2/30) (Table 2). The G5046A (V193I) polymorphism occurred in a patient with a grade II tumor (Table 2). 

### 3.3. Polymorphisms in the ND3 Gene

In *ND3*, there were only two sense-change polymorphisms (A10398G, T10084C), which occurred in a total of four patients (Appendix A). They did not influence the protein’s isoelectric point (Appendix A). According to the analysis in the SIFT Sequence program, these polymorphisms did not affect protein function (Appendix A), and they were not scored as pathogenic according to the APOGEE predictor (Appendix A). The missense polymorphism A10398G (T114A) occurred at a frequency of 10% (3/30) in the tested material, exclusively in patients diagnosed with stages II and III. The T10084C (I9T) polymorphism resulted in the changes in the amino acids forming the second helix. Its transmembrane segment began at position 2 and ended at position 24, while the correct positions should be 5 and 27, respectively (Appendix A Appendix A).

### 3.4. Polymorphisms in the ND4 and ND4L Genes

There were twelve types of polymorphisms in *ND4*, including one sense change T11253C. The most common polymorphism in the ND4 subunit gene in the study group was the synonymous change A11467G with a frequency of 26.6(6)% (8/30) (Table 2). There were only four synonymous polymorphisms in ND4L, with the most common *A10550G* occurring at a frequency of 10% (3/30) (Table 2). The frequency of the missense polymorphism T11253C (I165T) (*ND4*) according to mtDB turned out to be rare (10), as in the study material, where it detected in a patient with glioblastoma multiforme (Table 2).

### 3.5. Polymorphisms in the ND5 Gene

Eighteen different polymorphisms were found in *ND5.* Eight of them were missense (G13145A, G12406A, T12811C, G13889A, C14003T, G13604C, A13748G, G13759A) and ten were synonymous. The most common polymorphism in the study group was the synonymous change G12372A with a frequency of 26.6(6)% (8/30) (Table 2).

The missense polymorphism G13145A (S270N) occurred with a frequency of 10% (3/30) in patients with grade II and III gliomas. (Table 2). It did not have a significant effect on the protein’s biochemical properties (Appendix A). The G13759A (A475T) polymorphism caused a slight decrease in the aliphatic index, stability index and grand average of hydropathicity of the protein (Appendix A). It occurred in one patient with a grade III tumor (Table 2). In the PSSM assessment, alanine obtained −1 and threonine 6 points (Appendix A). The remaining six polymorphisms only occurred in individual patients diagnosed with glioblastoma (Table 2). None of them caused any significant changes in the biochemical protein’s properties. It should be noted that the C14003T and G13604C polymorphisms reached APOGEE predictor values quite similar to pathogenic changes (0.41) and involved conserved regions (6 and 8 points, respectively) (Appendix A). The G13604C (S423T) polymorphism did not cause significant changes in the biochemical properties of the protein, except for an increase in percentage of the twenty-first alpha helix from 0.61% to 1.13% (Appendix A). Only three (G13889A, A13748G, G13759A) of the eight reported polymorphisms did not involve the conserved regions (Appendix A). No changes in amino acid positions were shown in the TMHMM program for any of the detected missense polymorphisms.

### 3.6. Polymorphisms in the ND6 Gene

There were six different synonymous polymorphisms and one missense polymorphism (A14582G) in the *ND6* gene sequence. Of these, C14167T had the highest frequency (10%), occurring in three patients (Table 2). The missense polymorphism A14582G (V31A) resulted in a slight decrease in the aliphatic index, a grand average of hydropathicity and a slight increase in the percentage of the second alpha helix (Appendix A). The change involved a highly conserved region (8 points) (Appendix A). It did not result in any deviation in the helices from the correct sequence as assessed by TMHMM Server.

### 3.7. Mutations

In total, three types of mutations were found in the tested material, two of which were missense (G13096A, C13854A) and one nonsense (G4853A). Heteroplasmy in tumor tissue was involved in all detected mutations (Table 2).

The missense mutation G13096A (V254M), which occurred in *ND5*, did not change the isoelectric point, affecting the other parameters of the protein properties to a small extent (Appendix A) In the PSSM assessment, valine scored 7 points, and methionine scored 0 points (Appendix A). The C13854A mutation, which involved *ND5*, resulted in the appearance of the STOP codon (UAA) instead of tyrosine at position 506. This caused the isoelectric point of the protein to drop from 9.14 to 8.52 and the percentage of the twenty-third helix to decrease from 2.52% to 1.23%. In addition, there was an increase in the grand average of hydropathicity from 0.572 to 0.603 and also a slight decrease in the aliphatic index and protein stability (Appendix A). Using the TMHMM Server program, the amino acid sequence of the ND5 protein was shortened to end at position 508 in the mitochondrial matrix (Appendix A Appendix A). The evaluation of amino acid conservativeness and the influence of mutations in protein function are presented in Appendix A.

## 4. Discussion

Mutations in complex I that transport electrons from NADH to ubiquinone and pump protons outside of the inner mitochondrial membrane have been described in many cancers. In kidney and colorectal cancer, *ND3*, *ND4* and *ND5* were defined as hot spots, i.e., places with the highest risk of mutations in complex I [3]. In colon cancer and non-small cell lung cancer (NSCLC), polymorphisms found in *ND1* gene were associated with the presence of distant metastases [19]. The missense polymorphism T4216C (Y304H) has been detected in prostate cancer (1/16) [20] and in colorectal cancer at a frequency of 27% [21]. Transition of T4216C (Y304H) occurred in the tested material as a polymorphism in four patients. In the Polish population, this polymorphism was described in breast cancer patients (4/50) [7]. There are reports indicating its correlation with an increased risk of breast cancer, especially when it co-occurs with the A10398G change [22]. In kidney cancer, the T4216C polymorphism occurred as a somatic mutation in three patients [7]. This change has been found in patients with hereditary optic neuropathy (LHON) [23]. There are reports that the presence of T4216C in the blood increases the risk of insulin-resistant type 2 diabetes [24]. At protein position 304, histidine was slightly more preferred than tyrosine (9 vs. 6 PSSM points). It seems significant that the variant influenced the isoelectric point of the protein, changing it from 6.11 to 6.29, and caused the percentage of the thirteenth helix to drop from 8.32% to 4.86%. The change, however, did not affect a conserved region (3 points), which was determined in the SIFT Sequence program as harmless to the functioning of the protein. The change may contribute to a disruption of the body’s energy metabolism, especially since the APOGEE predictor indicates it as a pathogenic change. It is found in numerous subgroups of haplogroups characteristic of Europeans (K, R, H, X), but also of Africans (L1, L4) and Asians (A, D, M). The frequency of this variant in the population is quite high ((244)-frequency of occurrence from mtDB). It is difficult to clearly link it to the neoplastic process, the more so as the Uniprot database indicates Y304H as a naturally occurring variant in ND1 [25]. The occurrence of this transition in various diseases, including cancer, indicates that polymorphisms and mutations may reveal their phenotype depending on other accompanying factors, including other polymorphisms. It is possible that sense polymorphisms are the ones that contribute to this.

The somatic A3505G (T67A) mutation in the *ND1* region, described in prostate cancer [20] and pancreatic cancer cell lines [5], was identified as polymorphism in two of our patients. The variant defines the European haplogroup W and is also associated with mitochondrial subgroups characteristic for African (L2a1h, L4b1a, L3f1b4) and Asian (D1g5) countries. Threonine was the more preferred amino acid at protein position 67 than alanine (3 vs. 0 PSSM points). Thus, the change from medium-sized polar threonine to small hydrophobic alanine may be disadvantageous, especially since it involved a conserved region (7 points). All the more so because there has been a shift of amino acids from the mitochondrial matrix to the transmembrane segment and an elongation of the extra-mitochondrial part of the protein. Analysis in the SIFT Sequence program confirmed the detrimental effect of this variant on the functioning of the protein. The rarity of this change (56) in relation to the reference sequence (2648) also indicates the effect on the functioning of glial cells and the predisposition to tumor development. Polymorphism did not co-occur with any of the detected somatic mutations, but it was accompanied by a mutation in *IDH1* (R132H).

Another detected *ND1* missense polymorphism assessed in the SIFT Sequence program as detrimental to protein function is C3992T (T229M). The change is characteristic of the European mitochondrial subgroups (H3g1, H4, T2b4c) and is much less frequent (24) than the reference sequence 3992C (2680), indicating a possible association with brain tumors. In the Polish population, this polymorphism was described in patients with breast cancer, where it was detected in 1 of 50 patients [26]. Moreover, it has been described as a somatic mutation increasing the risk of thyroid cancer [5]. It is difficult to say what determines the potential harmfulness of this change on the functioning of mitochondria. In silico biochemical analysis showed a significant change in the stability index from 41.94 to 44.06, with the protein remaining unstable, as well as a significant increase in the percentage of the tenth alpha helix from 0.70% to 2.37%. Another ND1 polymorphism described in the Polish population in breast cancer patients (1/50), which, in the studied material, was detected in one patient with grade III tumor diagnosis, was A3796G (T164A) [7]. The relationship between this transition and a higher risk of developing dystonia in adults has also been described [18]. This polymorphism is very rare (10) in the population, opposed to the reference sequence 3796A (2677). Replacing a medium-sized polar threonine with a small hydrophobic alanine did not significantly affect the changes in the in silico evaluation of the biochemical properties of the protein. Additionally, this variant occurred in a highly variable region (1 point). 

The A4024G (T240A) polymorphism *ND1* defining the European H4a subgroup and the Asian M33d subgroup is a rare change (22) compared to the reference sequence 4024A (2682). Although this polymorphism was not associated with any specific diseases in the literature, it cannot be ruled out that it predisposes cells to tumorigenesis. Analysis in the SIFT Sequence program revealed the detrimental consequence of this transition on the functioning of the protein, which was confirmed by the unfavorable assessment of the conversion of medium-sized polar threonine to a small hydrophobic alanine (1 vs. 4 PSSM points). Position 240 in ND1 is close to average conservation (4 points). 

Another detected polymorphism *ND1* associated not only with European (W, X, U) but also with Asian (B, M) haplogroups is the A4093G (T263A) polymorphism, involving a region of near-medium conservation (4 points). The incidence of this change in the population is very rare (2), and in the examined material it was detected in one patient diagnosed with grade IV. Although according to the SIFT Sequence program it can be considered a change that is harmless to the functioning of the protein, this variant caused an increase in the percentage of the eleventh alpha helix from 1.95% to 3.17%. However, there are no reports of the occurrence of this polymorphism in specific disease entities.

In a study conducted on a Polish population, a total of 28 changes were detected in *MT-ND1*, *MT-ND2*, *MT-ND3* and *MT-ND6* in breast cancer cells, which were described in databases as polymorphisms. These changes were most common in the *MT-ND2* gene (50%, 14/28) and were not present in healthy cells [7]. Polymorphisms *ND2* G5460A (A331T) and G5046A (V193I), which were detected in our study material, occur in numerous mitochondrial subgroups characteristic of Europeans, Africans and Asians. No harmful effects of those polymorphism were shown in the conducted in silico analyses. However, there are reports indicating that the G5460A (A331T) variant is associated with the onset of Alzheimer’s disease [27]. The G5046A (V193I) polymorphism was described in breast cancer in the Polish population [28]. The change involves a conserved amino acid; therefore, it cannot be ruled out that may affect the glial neoplastic process.

One G4853A mutation was also detected in *ND2* and found to be synonymous. Mutations in mitochondrial DNA in healthy salivary glands of tobacco smokers were investigated, assuming that they could become a marker for the early stages of cancer development [29]. The G4853A mutation was also detected in 3/16 smokers. The change may contribute to oxidative damage to the mitochondrial genome [29]. If polycyclic aromatic hydrocarbons and nitrosamines in tobacco smoke are assumed to preferentially damage mtDNA, increasing the frequency of mutations therein, Hydrogen peroxides or hydroxyl radicals formed due to oxidative changes in mtDNA may, in turn, damage proteins, lipids or nuclear DNA (nDNA) [29]. A synonymous somatic mutation in the *ND2* region in glioblastoma (T4646C) was described by Kirches E. et al. [30]. A4613G and C4940T are synonymous changes described in the literature in patients with thyroid cancer [5]. The occurrence of synonymous changes indicates their role in the manifestation of other changes in the cell. They may influence how mutations in mtDNA or polymorphisms within it will manifest phenotypically.

One of the first cancer-related polymorphisms discovered was the transition of guanine to adenine at position 10398, causing a change in codon A114T in the *ND3* gene and disruption of the structure of complex I of the mitochondrial respiratory chain. African American women with this polymorphism have a higher likelihood of developing breast cancer. This polymorphism increases the likelihood of developing breast cancer in women with guanine at this position (G10398A). However, this relationship did not apply to Caucasian women [31]. A study conducted among the Polish population showed that the A10398G polymorphism was nevertheless associated with higher morbidity of breast cancer [32]. A similar conclusion was made in a study conducted on the population of European and American women with no Semitic ancestry [33]. In addition to the A10398G polymorphism, two others have been described in correlation with an increased risk of breast cancer: T16519C and G9055A [33], while the T3197C (OR = 0.31; CI 95%: 0.13–0.75; *p* = 0.0043) and G13708A (OR = 0.47; CI 95%: 0.24–0.92; *p* = 0.022) polymorphisms have been described as reducing this risk [32]. The G10398A polymorphism has also been described as increasing the risk of NSCLC and oral cancer [34]. There have also been reports of its association with hereditary optic neuropathy, metabolic syndrome and attention deficit hyperactivity disorder (ADHD) [35]. Moreover, it may be a protective factor against Parkinson’s disease [35]. The A10398G (T114A) variant occurs in a very large number of mitochondrial subgroups. It is most abundant in European, but also in African (L0, L1, L3) and Asian N haplogroups. In the examined material, it occurred in three patients with glioma, one with II degree and two with III degree. The change involved a moderately conserved region (4 points). According to the SIFT Sequence, the variant was considered harmless to the functioning of the protein. In patients with colorectal cancer, the A10398G polymorphism was considered to promote distant metastases [36]. However, no relationship was found between overall survival or progression-free survival in colorectal cancer and the A10398G polymorphism or five others: T479C (D-loop), T491C (D-loop), T10035C (*MT-TG*), A13781G (*MT-ND5*) and T16189C (D-loop) [36]. Despite these reports in the literature, it is difficult to associate the A10398G variant with brain gliomas based on the results of the present study. Another *ND3* polymorphism T10084C (I9T) detected in the material tested is found in very numerous mitochondrial subgroups characteristic of Europeans, Africans and Asians. It has not been described in cancer so far. At this position of the protein, threonine was as preferred as isoleucine, and the change did not involve a conserved region. The effect on the functioning of mitochondria may be demonstrated by the shift of amino acids within the transmembrane segment of the first alpha helix (assessed in the TMHMM program) and a decrease in its percentage from 53.87% to 40.47%. In the SIFT Sequence program, it was rated as harmless to the functioning of the protein.

One missense polymorphism (T11253C) was detected in *ND4*. In the population, the T11253C variant is rare (10), opposed to the reference sequence (2694) (I165T), and is characteristic of the L4a mitochondrial subgroup associated with the African population, but also H6a1a and H35a for Caucasians. Although the T11253C (I165T) variant in the assessment of the SIFT Sequence program is neutral, it should be noted that the replacement of medium-sized hydrophobic isoleucine with medium and polar threonine concerned a region with high conservation (6 points) and, in the in silico analysis of biochemical properties of the protein, the change decreased by almost half the percentage of the eighth helix from 2.56% to 1.16%, which may indicate a change affecting respiratory chain function. The rarity of this change in the European population and the influence on biochemical properties may point to its association with the neoplastic process.

In the tested material, the most polymorphisms (8) were detected in *ND5*. Hayashi’s group, in their research into the role of mtDNA in carcinogenesis, demonstrated mutagenicity of the G13997A and 13885.1C transitions in *ND5* [36]. Both of these changes were responsible for respiratory complex I dysfunction and increased production of ROS in the A11 line of lung cancer and P82M for fibrosarcoma which correlated with an increased risk of metastasis [36]. Another ND5 polymorphism, namely, G13759A (A475T), was detected in the Polish population in breast cancer [27]. In the study material, it was detected in one patient with grade III tumor diagnosis.

In the material tested, the G13604C (S423T) polymorphism was detected in a patient with grade IV tumor diagnosis, which has not been reported in the literature so far. It is also missing from mitochondrial databases. This polymorphism involved an amino acid with high conservation (8 points). Although this variant was not assessed as detrimental to protein function according to the SIFT program, the threshold value it obtained in APOGEE, an increase of almost half the percentage of the twenty-first alpha helix (0.61% vs. 1.13%), and the PSSM score may indicate its influence on the functioning of the respiratory chain and its association with the neoplastic process.

G12406A (V24I) polymorphism of *ND5* was found in numerous mitochondrial subgroups belonging to haplogroups characteristic of Europeans (X, T, R, U), Africans (L2, L3) and Asians (M, F, P). Transition occurred in only one patient with a degree IV tumor diagnosis. Although the in silico biochemical analysis and the SIFT Sequence evaluation did not confirm the harmfulness of the change, it cannot be ruled out that it may be related to carcinogenesis. It should be noted that it involves a conserved region (5 points), and the computer analysis according to APOGEE indicated a pathogenic change. This variant has been detected in breast cancer and in gastric cancer cells [37], which would indicate a possible relationship with the neoplastic process.

Noteworthy polymorphism found in *ND5* is A13748G (N471S). It did not involve a conserved region, but it is characteristic of the F1a3a subgroup found in Asia. It was found in a patient with glioblastoma. This is a very rare change (2) compared to the reference sequence (2702). There are no reports in the literature linking it to cancer. However, the G13759A (A475T) polymorphism has been described in the literature in 2 patients with glioblastoma among 32 patients studied, as well as in patients with colon adenocarcinoma (2/86) [38]. In the material tested, the G13759A polymorphism was found in one patient with grade III tumor diagnosis. This variant is found in a large number of mitochondrial subgroups. The A475T change is identified in the Uniprot database as a naturally occurring variant in the ND5 subunit [24]. However, it should be noted that both polymorphisms involve closely spaced codons (N471S and A475T), which indicates that changes in this region may somehow be preferential for cancer cells.

In the *ND6* study on a population of 50 Polish breast cancer patients, only synonymous polymorphisms were detected [7]. In the analyzed material, one polymorphism of the sense change A14582G (V31A) was detected, which is characteristic for the H4a subgroup present among Caucasians. This is a rather rare change (23) compared to the reference sequence (2681). Although the change did not affect the functioning of the protein according to the SIFT Sequence program, it involved a highly conserved region (8 points), and valine is the preferred amino acid at this position. In the literature, it has been described in LHON [39]; it has not yet been described in cancer.

It is assumed that not only missense change in the mitochondrial DNA may affect mitochondrial function. Polymorphisms or mutations that do not change the protein are often found in a variety of disease entities, including cancer. Studies on bacterial mitochondria have confirmed that synonymous mtDNA variants may interfere with the protein translation process. Four synonymous changes were detected in *ND1* (A3480G, G3915A, T3826C) that were not described in the literature on cancer. The most common synonymous polymorphism was the A4769G change, which occurred with a frequency of 100% (30/30) in *ND2*. It is characteristic of numerous mitochondrial subgroups. The change has been described in breast cancer [7,28]. Another synonymous polymorphism reported to increase the risk of age-related macular degeneration is the A4917G polymorphism [40]. In the Polish population, the T5004C polymorphism has been found in patients with breast cancer [26]. It occurs in mitochondrial subgroups characteristic of Asians (D1j1a, D2b1a), but also Africans (L5a1a) and Europeans (H4). The most common polymorphism in *ND4* in the study group was the synonymous A11467G change with a frequency of approx. 27% (8/30). There are reports of a relationship between this polymorphism and the incidence of oral cancer in the Indian population [41]. The change is characteristic of the U haplogroup and the T3 subgroup, which are present among populations of European descent.

Two heteroplasmic missense mutations (G13096A, C13854A) in the *ND5* region were detected in the material tested. They occurred individually in two (2/30) patients with glioblastoma (IV). The G13096A (V254M) transition involved a highly conserved region (9 points) and was assessed as detrimental to the protein functioning, scoring 0.04 points in the SIFT Sequence program. However, according to the APOGEEC predictor, this transition was assessed as neutral with a value of 0.31 points and a cut-off point of 0.5. In the case of the C13854A mutation, a STOP UAA codon occurred instead of tyrosine at protein position 506. This is a nonsense mutation. The position of the protein where the change occurred was a variable region (1 point). The occurrence of a stop codon interrupts further amino acid addition and a shorter ND5 protein is formed. Therefore, it is plausible that the change had an impact on the functioning of mitochondria, especially since it influenced the biochemical properties of the protein assessed in silico. The heteroplasmy of the detected mutations and the sites involved suggest their pathogenic nature, indicating them as the cause or result of glial neoplasms. Both mutations are missing from the mtDNA–Human Mitochondrial Genome Database and have been not described in the literature so far.

## 5. Conclusions

Polymorphisms A3505G, C3992T, A4024G, T4216C, G5046A, G7444A, T11253C, G12406A, G13604C and mutations G13096A and C13854A are changes that may have an effect on the functioning of mitochondria by affecting the biochemical properties of proteins, and they have been assessed in silico as deleterious, indicating their association with carcinogenesis. The A13748G polymorphism that belongs only to the mitochondrial subgroup found outside the Caucasian population (F1a3a) may predispose to the incidence of brain tumors in this race. Polymorphisms that are rarely found in databases and described in other diseases, including neurological diseases, may predispose to cancer, including brain gliomas.

## Figures and Tables

**Table 1 biomedicines-11-01183-t001:** Clinical characteristics of patients.

Patient Number	Sex	Age	Histopathological Diagnosis	Mutation *IDH1*	Mutation *IDH2*	The Size of the Tumor (mm)	The Location of the Tumor
Grade IV							
1	M	51	Glioblastoma multiforme with poorly differentiated component	-	-	34 × 31 × 26	right parietal lobe
2	F	57	Glioblastoma multiforme	-	-	44 × 36 × 30	right temporal lobe
3	M	49	Glioblastoma multiforme	R132H (c. 395 G>A)	R172K (c. 515 G>A)	24 × 34 × 35	right frontal lobe
4	M	51	Glioblastoma multiforme	-	-	33 × 30 × 26	the right frontal–eclipse area
5	F	68	Glioblastoma multiforme	-	-	45 × 34 × 36	right occipital lobe
6	M	52	Glioblastoma multiforme	-	-	70 × 51 × 60	left temporal lobe
7	M	66	Glioblastoma multiforme	-	-	52 × 44 × 47	right temple–left frontal–parietal area
8	M	58	Glioblastoma multiforme	-	-	60 × 45 × 50	right temporal lobe
9	F	43	Glioblastoma multiforme with a giant cell component	-	-	62 × 57 × 46	left frontal lobe
10	F	66	Glioblastoma multiforme	-	-	51 × 45 × 34	right occipital lobe
11	M	46	Glioblastoma multiforme	-	-	32 × 30 × 31	left frontal lobe
12	F	56	Glioblastoma multiforme	-	-	36 × 21 × 19	right temporal lobe
13	M	69	Glioblastoma multiforme	-	-	80 × 70 × 70	right temporal lobe
14	F	66	Glioblastoma multiforme with a primitive neuronal component	-	-	60 × 40 × 55	left parietal lobe
15	M	76	Glioblastoma multiforme	-	-	48 × 40 × 38	left parietal lobe
16	M	71	Glioblastoma multiforme with a primitive neuronal component	-	-	46 × 47 × 36	left temporal lobe
17	M	48	Glioblastoma multiforme	-	-	34 × 30 × 25	left parietal–occipital area
Grade III							
18	F	40	Pilocytic astrocytoma with anaplasia	-	-	33 × 29 × 25	right temporal lobe
19	M	39	Anaplastic astrocytoma	-	-	23 × 20 × 28	right temporal lobe
20	F	48	Anaplastic astrocytoma	R132H (c. 395 G>A)	-	54 × 51 × 52	the right occipital–parietal area and the body of the corpus callosum
21	M	68	Anaplastic astrocytoma	-	-	45 × 39 × 35	the right frontal–eclipse area
22	F	34	Anaplastic astrocytoma	-	-	61 × 56 × 68	right frontal lobe
23	M	37	Anaplastic oligodendroglioma	-	-	74 × 64 × 50	right temporal–frontal–parietal area
Grade II							
24	M	22	Astrocytoma gemistociticum	-	-	28 × 25 × 16	right temporal lobe
25	F	32	Astrocytoma diffusum	R132H (c. 395 G>A)	-	21 × 20 × 14	left temporal–frontal–parietal area
26	M	43	Astrocytoma diffusum	-	-	50 × 40 × 35	right frontal lobe
27	F	41	Oligodendroglioma	-	-	20 × 15 × 10	right frontal lobe and frontal–parietal area
28	M	52	Astrocytoma diffusum	-	R172S (c. 516 G>T)	60 × 70 × 60	right frontal lobe
29	M	29	Oligodendroglioma	-	-	28 × 20 × 15	right frontal lobe
30	M	28	Oligodendroglioma	-	-	50 × 40 × 35	right parietal–occipital area

*IDH*—gene of isocitrate dehydrogenase; F—female; M—male.

**Table 2 biomedicines-11-01183-t002:** Summary of all detected polymorphisms and mutations in genes of the mitochondrial complex I.

**Polymorphism**	**The Number of Patients with Diagnosed Polymorphism with the Diagnosis of Tumor Grade**			**The Number of All Patients in Whom Polymorphism Was Detected**	**Region**	**Amino Acid Change**	**Type of Change**	**Described in the Available Literature**
	**II**	**III**	**IV**					
A3480G	1/7	2/6	0/17	3/30	*MT-ND1*	L ^58^ L	synonymous	yes
**A3505G**	**1/7**	**0/6**	**1/17**	**2/30**	** *MT-ND1* **	**T ^67^ A**	**missense**	**yes**
**A3796G**	**0/7**	**1/6**	**0/17**	**1/30**	** *MT-ND1* **	**T ^164^ A**	**missense**	**yes**
T3826C	1/7	0/6	0/17	1/30	*MT-ND1*	L ^174^ L	synonymous	yes
G4727A	0/7	0/6	1/17	1/30	*MT-ND1*	G ^203^ G	synonymous	yes
**C3992T**	**0/7**	**0/6**	**1/17**	**1/30**	** *MT-ND1* **	**T ^229^ M**	**missense**	**yes**
**A4024G**	**0/7**	**0/6**	**1/17**	**1/30**	** *MT-ND1* **	**T ^240^ A**	**missense**	**yes**
**A4093G**	**1/7**	**0/6**	**0/17**	**1/30**	** *MT-ND1* **	**T ^263^ A**	**missense**	**yes**
**T4216C**	**1/7**	**1/6**	**2/17**	**4/30**	** *MT-ND1* **	**Y ^304^ H**	**missense**	**yes**
G4580A	0/7	0/6	1/17	1/30	*MT-ND2*	M ^37^ M	synonymous	yes
T4646C	1/7	0/6	1/17	2/30	*MT-ND2*	Y ^59^ Y	synonymous	yes
A4917G	1/7	1/6	0/17	2/30	*MT-ND2*	N ^150^ D	synonymous	yes
A4727G	0/7	0/6	1/17	1/30	*MT-ND2*	M ^86^ M	synonymous	yes
A4769G	7/7	6/6	17/17	30/30	*MT-ND2*	M ^100^ M	synonymous	yes
A4793G	0/7	0/6	1/17	1/30	*MT-ND2*	M ^108^ M	synonymous	yes
T5004C	0/7	0/6	1/17	1/30	*MT-ND2*	L ^179^ L	synonymous	yes
**G5046A**	**1/7**	**0/6**	**0/17**	**1/30**	** *MT-ND2* **	**V ^193^ I**	**missense**	**yes**
G5147A	1/7	0/6	1/17	2/30	*MT-ND2*	T ^226^ A	synonymous	yes
T5201C	1/7	0/6	0/17	1/30	*MT-ND2*	I ^244^ I	synonymous	yes
**G5460A**	**1/7**	**0/6**	**1/17**	**2/30**	** *MT-ND2* **	**A ^331^ T**	**missense**	**yes**
T5495C	1/7	0/6	0/17	1/30	*MT-ND2*	F ^342^ F	synonymous	yes
**T10084C**	**0/7**	**0/6**	**1/17**	**1/30**	** *MT-ND3* **	**I ^9^ T**	**missense**	**yes**
**A10398G**	**1/7**	**2/6**	**0/17**	**3/30**	** *MT-ND3* **	**T ^114^ A**	**missense**	**yes**
C10793T	0/7	0/6	1/17	1/30	*MT-ND4*	L ^12^ L	synonymous	yes
C11191T	0/7	0/6	1/17	1/30	*MT-ND4*	N ^144^ N	synonymous	yes
A11251G	1/7	1/6	0/17	2/30	*MT-ND4*	L ^164^ L	synonymous	yes
**T11253C**	**0/7**	**0/6**	**1/17**	**1/30**	** *MT-ND4* **	**I ^165^ T**	**missense**	**yes**
T11299C	1/7	2/6	0/17	3/30	*MT-ND4*	T ^180^ T	synonymous	yes
G11377A	1/7	1/6	0/17	2/30	*MT-ND4*	K ^206^ K	synonymous	yes
A11467G	2/7	4/6	2/17	8/30	*MT-ND4*	L ^236^ L	synonymous	yes
A11470G	0/7	1/6	0/17	1/30	*MT-ND4*	K ^237^ K	synonymous	yes
C11674T	0/7	0/6	1/17	1/30	*MT-ND4*	T ^305^ T	synonymous	yes
A11812G	1/7	1/6	0/17	2/30	*MT-ND4*	L ^351^ L	synonymous	yes
T11899C	0/7	0/6	1/17	1/30	*MT-ND4*	S ^380^ S	synonymous	yes
G11914A	0/7	1/6	0/17	1/30	*MT-ND4*	K ^385^ K	synonymous	yes
A11947G	0/7	0/6	1/17	1/30	*MT-ND4*	T ^396^ T	synonymous	yes
A10550G	1/7	2/6	0/17	3/30	*MT-ND4L*	M ^27^ M	synonymous	yes
G10646A	1/7	0/6	0/17	1/30	*MT-ND4L*	V ^59^ V	synonymous	yes
A10679G	0/7	0/6	1/17	1/30	*MT-ND4L*	E ^70^ E	synonymous	yes
C10698T	1/7	0/6	0/17	1/30	*MT-ND4L*	L ^77^ L	synonymous	yes
G12372A	2/7	4/6	2/17	8/30	*MT-ND5*	L ^12^ L	synonymous	yes
**G12406A**	**0/7**	**0/6**	**1/17**	**1/30**	** *MT-ND5* **	**V ^24^ I**	**missense**	**yes**
T12414C	1/7	0/6	0/17	1/30	*MT-ND5*	P ^26^ P	synonymous	yes
C12705T	1/7	0/6	0/17	1/30	*MT-ND5*	I ^123^ I	synonymous	yes
**T12811C**	**1/7**	**0/6**	**1/17**	**2/30**	** *MT-ND5* **	**Y ^159^ H**	**missense**	**yes**
C12858T	0/7	0/6	1/17	1/30	*MT-ND5*	Y ^174^ Y	synonymous	yes
**G13145A**	**1/7**	**2/6**	**0/17**	**3/30**	** *MT-ND5* **	**S ^270^ N**	**missense**	**yes**
T13617C	1/7	1/6	1/17	3/30	*MT-ND5*	I ^427^ I	synonymous	no
G13368A	1/7	1/6	0/17	2/30	*MT-ND5*	G ^344^ G	synonymous	yes
**G13604C**	**0/7**	**0/6**	**1/17**	**1/30**	** *MT-ND5* **	**S ^423^ T**	**missense**	**no**
T13743C	1/7	0/6	0/17	1/30	*MT-ND5*	T ^469^ T	synonymous	yes
**A13748G**	**0/7**	**0/6**	**1/17**	**1/30**	** *MT-ND5* **	**N ^471^ S**	**missense**	**yes**
**G13759A**	**0/7**	**1/6**	**0/17**	**1/30**	** *MT-ND5* **	**A ^475^ T**	**missense**	**yes**
A13866G	1/7	0/6	0/17	1/30	*MT-ND5*	K ^509^ K	synonymous	no
**G13889A**	**0/7**	**0/6**	**1/17**	**1/30**	** *MT-ND5* **	**C ^518^ Y**	**missense**	**yes**
T13896C	0/7	1/6	0/17	1/30	*MT-ND5*	F ^520^ F	synonymous	yes
**C14003T**	**0/7**	**0/6**	**1/17**	**1/30**	** *MT-ND5* **	**T ^556^ I**	**missense**	**yes**
G14040A	0/7	1/6	0/17	1/30	*MT-ND5*	Q ^568^ Q	synonymous	yes
C14167T	1/7	2/6	0/17	3/30	*MT-ND6*	E ^169^ E	synonymous	yes
T14182C	0/7	0/6	1/17	1/30	*MT-ND6*	V ^164^ V	synonymous	yes
A14233G	1/7	1/6	0/17	2/30	*MT-ND6*	D ^147^ D	synonymous	yes
G14305A	0/7	0/6	1/17	1/30	*MT-ND6*	S ^123^ S	synonymous	yes
C14365T	0/7	0/6	1/17	1/30	*MT-ND6*	V ^103^ V	synonymous	yes
C14620T	1/7	0/6	1/17	2/30	*MT-ND6*	G ^18^ G	synonymous	yes
**A14582G**	**0/7**	**0/6**	**1/17**	**1/30**	** *MT-ND6* **	**V ^31^ A**	**missense**	**yes**
**Sum of detected changes**	42	40	55	136	-	-	-	-
**Mutation**	**The number of patients who have been diagnosed with a mutation with a diagnosis of tumor grade**			**Number of all patients in whom the mutation was detected**	**Region**	**Amino acid change**	**Type of change**	**Described in the available literature**
	**II**	**III**	**IV**					
G4853A	0/7	0/6	1/17 ^(1)^	1/30	*MT-ND2*	L ^128^ L	synonymous	yes
**G13096A**	**0/7**	**0/6**	**1/17 ^(2)^**	**1/30**	** *MT-ND5* **	**V ^254^ M**	**missense**	**no**
C13854A	0/7	0/6	1/17 ^(3)^	1/30	*MT-ND5*	Y ^506^ STP	codon STOP (UAA)	no
**Sum of detected changes**	0		3	3/30	-	-	-	-

Heteroplasmia: ^(1)^ tumor: A—56.1%, G—43.9%; ^(2)^ tumor: A—69.1%, G—30.9%; ^(3)^ tumor: C—65.8%, T—34.2%. Missense changes are marked in bold. Abbreviations: *MT-ND1*, *MT-ND2*, *MT-ND3*, *MT-ND4*, *MT-ND4L*, *MT-ND5*, *MT-ND6*—genes encoding ND1, ND2, ND3, ND4, ND4L, ND5, ND6 subunits in the respiratory chain complex; L—leucine; V—valine; T—threonine; A—alanine; G—glycine; M—methionine; Y—tyrosine; H—histidine; N—asparagine; D—aspartic acid; I—isoleucine; F—phenylalanine; L—leucine; K—lysine; S—serine; P—proline; C—cysteine; Q—glutamine; E—glutamic acid; D—aspartic acid; L—leucine; V—valine; M—methionine; Y—tyrosine; STP—anticodon stop; *MT-ND2*, *MT-ND5*—genes encoding ND2, ND5.

## Data Availability

All data are included in the article and Appendix A.

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
