# Peer review of "Mitochondrial DNA Changes in Respiratory Complex I Genes in Brain Gliomas"

_biomedicines, 2023, doi:10.3390/biomedicines11041183_

Round 1

Reviewer 1 Report

The authors analyze mitochondria DNA sequences from thirty patients with brain gliomas. The samples include brain tumors and peripheral blood to determine the polymorphism or mutation sequence. They identify many polymorphisms in ND1-ND6 genes and three mutations in ND2 and ND5 genes. They also describe the biochemical properties and possible functional impacts on some missense changes. This study is a valuable reference for brain gliomas research and provides the information for further mitochondria complex I biochemical and functional investigation.

I have a minor question: what is the meaning of the numbers of reference sequences in the discussion, such as 3796A (2677), 4024A (2682), and (2702)?

Author Response

Dear

Reviewer

We would like to thank You for your generous comments on the manuscript. The response to your comments is listed below this letter With regards

Ludmiła Grzybowska-Szatkowska

Paulina Kozakiewicz

Reviewer

The authors analyze mitochondria DNA sequences from thirty patients with brain gliomas. The samples include brain tumors and peripheral blood to determine the polymorphism or mutation sequence. They identify many polymorphisms in ND1-ND6 genes and three mutations in ND2 and ND5 genes. They also describe the biochemical properties and possible functional impacts on some missense changes. This study is a valuable reference for brain gliomas research and provides the information for further mitochondria complex I biochemical and functional investigation.

I have a minor question: what is the meaning of the numbers of reference sequences in the discussion, such as 3796A (2677), 4024A (2682), and (2702)?

  • It’s from mtDB-Human Mitochondrial Genome Database on site http://www.mtdb.igp.uu.se/ - the frequency of occurrence of the change sequence or the frequency of occurrence of the reference sequence .It is from table 5 as we’ve cited in results.

We added at the beginning of the discussion: [(244)- frequency of occurrence from mtDB]

Reviewer 2 Report

i read with great interest the article by Kozakiewicz et al titled "Mitochondrial DNA changes in respiratory complex I genes in brain gliomas" where the authors evaluated the role of mitochondria in the process of carcinogenesis.  the study collected blood material from 30 patients diagnosed with glioma II, III, and IV grade. 

also, the changes in the genes of complex I were assessed in terms of the polymorphisms via transcriptomic utilizing next-generation sequencing performed on the MiSeqFGx apparatus (Illumina).

there are a few comments:

1-The paper is well-written and has an excellent study design, however, the statistical methods should be elaborated as it is not discussed. In addition, power analysis should be included to justify the 30 patients involved.

2-Complete patient demographics and clinical data should be included

3-A copy of the IRB should be included in the supplementary data.

4-the discussion is very long ad needs to be shortened

5- the results should be further explained and evaluated via systems biology to study pathways if possible.

Author Response

Dear                                                                                             

Reviewer

We would like to thank You for  your generous comments on the manuscript. We improved the manuscript by adding and removing some fragments. We can not attach a copy of IRB to supplementary material. In the IRB, there are signatures of the bioethics committee, these are personal data that are subject to protection (RODO). We may attach a copy of IRB only for reviewers to view and not in the supplementary material.

Point-by-point responses to reviewer comments are listed below this letter

With regards

Ludmiła Grzybowska-Szatkowska

Paulina Kozakiewicz

1-The paper is well-written and has an excellent study design, however, the statistical methods should be elaborated as it is not discussed. In addition, power analysis should be included to justify the 30 patients involved

– In the case of a brain tumor, it is difficult to obtain histopathological material for comparison with healthy brain tissue. In addition, obtaining 30 cases of brain tumors is also difficult to obtain due to the rarity of the tumor. Only bioinformatic programs were used in the article. The group of patients and the assumption of the work did not require the use of statistical programs.

2-Complete patients demographics and clinical data should be included

  • We have attached information about the sex of the patients to the table 1. The aim of the study was not to assess the relationship between the effectiveness of the used therapies and the occurrence of polymorphisms. More clinical data on patients is not required for this article. All patients had undergone surgery to remove the tumor, during which material for research has been obtained. As we included in the material section, patients had not received any treatment before surgery. Further treatment of the patient did not affect the obtained results.

3-A copy of the IRB should be included in the supplementary data.

We may attach a copy of IRB only for reviewers to view and not in the supplementary material. In the IRB, there are signatures of the bioethics committee, these are personal data that are subject to protection (RODO)

4-the discussion is very long ad needs to be shortened

  • We shortened the discussion by one page. –from 3789 to 3116 words

5- the results should be further explained and evaluated via systems biology to study pathways if possible.- We used a bioinformatics program to evaluate the consequences of missense changes. There are no possibilities to evaluate via systems biology study pathways. Complex I belongs to the electron transport chain. We can present only a figure of this if it is necessary.
